# Age-Related Associations of Altruism with Attitudes towards COVID-19 and Vaccination: A Representative Survey in the North of Italy

**DOI:** 10.3390/bs13020188

**Published:** 2023-02-19

**Authors:** Verena Barbieri, Christian J. Wiedermann, Stefano Lombardo, Barbara Plagg, Giuliano Piccoliori, Timon Gärtner, Adolf Engl

**Affiliations:** 1Institute of General Practice and Public Health, Claudiana College of Health Professions, 39100 Bolzano, BZ, Italy; 2Department of Public Health, Medical Decision Making and Health Technology Assessment, University of Health Sciences, Medical Informatics and Technology, 6060 Hall, Austria; 3Provincial Institute for Statistics of the Autonomous Province of Bolzano—South Tyrol (ASTAT), 39100 Bolzano, BZ, Italy; 4Faculty of Education, Free University of Bolzano, 39100 Bolzano, BZ, Italy

**Keywords:** SARS-CoV-2, pandemic, altruism, vaccination, vaccine hesitancy, health behavior, social preferences

## Abstract

Background: During the coronavirus pandemic, altruism has been linked to personal protective behavior, vaccine development, and vaccination intention. Studies of the moderating effects of age on altruism in pandemic preparedness have not yet been conducted. Methods: A representative cross-sectional survey of residents of South Tyrol, Italy, was conducted in March 2021. Among the participants, 1169 were aged 18–69 years, and 257 were aged ≥ 70 years. The questionnaire collected information on sociodemographic and individual characteristics, including comorbidities, COVID-19-related experiences, trust in information, the likelihood of accepting the national vaccination plan, and altruism. A linear regression analysis was performed. Results: Among 1426 participants, the median altruism sum score was 24 (interquartile range, 20–26). In the participant group aged ≥ 70 years, the median altruism score was significantly higher than that in the younger group. Participants living in a single household were significantly less altruistic than other participants, while participants working in the health sector, living in a household at risk from coronavirus disease 2019, or suffering from a chronic disease were found to be more altruistic. Altruism showed significant positive correlations with age and agreement with the national vaccination plan and was negatively correlated with well-being. Trust in institutions was positively correlated with altruism only in the younger age group but not in the elderly. Linear regression models confirmed female gender and identified trust in institutions as a positive predictor of altruism. In the younger age group, increased well-being and restricted individual sports activities were associated with reduced altruism, whereas support of compulsory self-isolation after contact with a SARS-CoV-2-positive person and handwashing as a personal protective measure were positively associated. Conclusion: Altruism is associated with various predictors of pandemic behavior and traits. The strengths of the identified positive and negative correlations support the modifying role of age in the effects of altruism on pandemic attitudes. Interventions that are likely to enhance altruism to improve pandemic preparedness in certain age groups require further study.

## 1. Introduction

Altruism refers to unselfishness, selflessness, and a way of thinking and acting characterized by consideration for others [1]. It is not linked to an immediate benefit or countervalue; ultimately, the benefit of the actor is greater than the costs expended with altruistic behavior as a voluntary act of the actor [2]. Because people can value more than their own well-being, empathic concern for a person in need can lead to altruistic motivation that ultimately aims to improve that person’s well-being [3]. Altruism triggered by empathy may or may not always be good. In certain non-trivial circumstances, it may even pose a greater threat than selfish egoism [4].

Preventing disease through vaccination is one of the greatest achievements of medicine and has been successfully demonstrated in the coronavirus disease 2019 (COVID-19) pandemic [5]. Altruism is important for understanding individual vaccination decisions [6]. Altruism has been cited as an important factor in the success of Severe Acute Respiratory Syndrome *Coronavirus-2* (SARS-CoV-2) vaccine development because of the participation of volunteers in clinical trials [7]. Voluntary reporting of the side effects of new vaccinations is based on a large altruism component [8], and altruism is therefore important in vaccination surveillance. Using the number of organ transplants as a proxy for health altruism, since the concept of altruism forms the basis of transplant ethics [9], a direct relationship between transplant rates as a proxy variable and vaccination rates was found in Union’s countries during the COVID-19 pandemic, in which the decision to be vaccinated depended on the individual’s choice, confirming that altruism may impact vaccine uptake [10].

Mask-wearing, social distancing, and hand washing were recommended as three important factors to prevent the spread of the virus, and thus as altruistic habits [11]. Mask-wearing and vaccination lead to the same prosocial behavior [12] that, in the actual pandemic, was also found to be related to well-being [13].

Altruism during the pandemic was examined in a representative study in Germany for the age group of adults from 18 to 69 years [14]. Higher levels of altruism were associated with being female, being younger, having children, engaging in sports activities, having at least one chronic disease, and vaccination against COVID-19. According to Grimalda et al. [15], altruism increased in groups of people who were more exposed to COVID-19, altruism had a local character, and in COVID-19, altruism was regarded as more closed relative to other vulnerable people [16]. A low level of altruism was among the individual characteristics behind hesitant or dismissive vaccination behaviors, such as high levels of self-interest, impulsivity, argumentativeness, emotional instability, and reduced conscientiousness [17,18,19]. In a study of sexual and gender minority men and transgender women, higher levels of altruism were associated with increased willingness to be vaccinated [20].

Using a representative survey of residents of South Tyrol, we examined the demographic, social, and individual attitude correlates of hesitance to vaccinations, including altruism, and reported on the population characteristics of vaccine hesitancy; however, they did not find a significant effect of altruism on vaccine hesitancy [21]. Altruism has a positive effect on vaccine hesitancy in the rural population of South Tyrol [22]. Our aim was to investigate altruism in detail in light of the pandemic’s understanding of how altruism and pandemic-related aspects of vaccination are associated. Altruistic behavior increases with age, whereas antisocial tendencies decrease [23], so age may be an important moderating effect in the relationship between altruism and vaccination. Therefore, the present analysis addresses the influence of age on altruism-related pandemic attitudes.

## 2. Methods

The aim of the research was to investigate the extent to which altruistic behaviors are associated with health attitudes during a pandemic. A particular focus was on the question if individuals who exhibit more altruistic behaviors are more likely to engage in behaviors such as getting vaccinated. The methods used to address these issues involved the use of a survey research design to gather data from a large sample of individuals about their attitudes and behaviors. The survey includes questions about altruistic behaviors, public health behaviors, and demographic information. Consensus-based checklist recommendations for the reporting of survey studies (CROSS) were followed [24].

### 2.1. Study Design and Data Collection

Data were obtained from an extended version of the COSMO survey [25] performed in South Tyrol in March 2021, using a probability-based mode survey of participants aged > 17 years. The intelligibility and validity of the COSMO questionnaire were discussed using a published survey tool [26]. The survey was conducted in March 2021. Details of the study design have been previously reported [21,22], including descriptions of the recruitment of a random sample and sample size determination. South Tyrol is the northernmost Italian region (Province of Bolzano), with a catchment area of 531,178 multilingual but predominantly German-speaking inhabitants, characterized by an autonomous political system.

### 2.2. Altruism

Altruism was quantified using the ‘Elderly Care Research Center Altruism Scale’, a brief, reliable, and valid altruism scale useful for assessing this important prosocial orientation and resource among older adults and other age groups [27]. Answers on a 6-point Likert scale from 1 = “don’t agree at all” to 6 = “completely agree” were obtained for the statements (Altruism 1) “I enjoy doing things for others”, (Altruism 2) “I try to help others, even if they do not help me”, (Altruism 3) “Seeing others prosper makes me happy”, (Altruism 4) “I really care about the needs of other people”, and (Altruism 5) “I come first and should not have to care so much for others.” To aid in interpretation, responses to the five Altruism items were reverse-coded to compare them with other statements. Strong correlations among the altruism scale, salient personality traits, psychological well-being, religiosity, and meaning in life establish construct validity [27]. The items were summed to a sum score that was examined in detail and could take values from 5 to 30, where 5 indicated the lowest and 30 indicated the highest level of altruism.

### 2.3. Questionnaire

Sociodemographic data were collected to predict vaccine hesitancy. Vaccine hesitancy was measured using the dichotomous question, “Would you get vaccinated against COVID-19?”. The items covered trust in vaccination, beliefs about the COVID-19 vaccine itself, and opinions on COVID-19 vaccination.

Trust in information sources and institutions [28,29] (health authorities and politics) was measured on a 6-point Likert scale from “no trust” to “big trust”, and a seventh item, “don’t know”, was investigated as well as conspiracy perceptions (5 questions on a 6-point Likert scale from “don’t agree at all” to “completely agree”) [30], resilience (3 items on a 6-point Likert scale from “don’t agree at all” to “ completely agree”) [31], and well-being within the last 2 weeks (5 items on a 4-point Likert scale from “always” to “never” [32]). The sum of these variables is considered a potential predictor of altruism during the COVID-19 pandemic. Agreement with decision making during the pandemic regarding restrictions and vaccination was asked with a single question [17]. Agreement was asked on a 6-point Likert scale from 1 (don’t agree at all) to 6 (completely agree) and a seventh item “I don’t know was they have decided”. Agreement with the national vaccination plan was assessed using a single question on a 6-point Likert scale ranging from 1 (do not agree at all) to 6 (completely agree). Questions about their own behavior concerning prevention measures were taken from SteelFisher et al. [33]. The 10 questions were answered on a 6-point Likert scale from 1 (never) to 6 (always) and an additional seventh option “I don’t know”. Support for actual restrictions was assessed using 18 questions that could be answered on a 6-point Likert scale ranging from 1 (I support it completely) to 6 (I do not support it at all). Responses were recorded from 1 (I do not support it at all) to 6 (I support it completely).

### 2.4. Age Groups

The altruism scale used in our questionnaire was developed, especially for older persons up to 70 [18]. It is a valid and reliable instrument to measure altruism in this age group to be used in younger age groups as well [27]. Furthermore, we compared the effects of the predictors of altruism identified by Hajek and König [14], who used a different altruism scale. Since Hajek and König referred to individuals aged 18–70 years, we decided to group our analyses separately for the younger age group from 18 to 69 years, and for the older age group up to 70 years. Thus, the altruism scale used here has tested validity for the older age group and may have comparability with data from Hajek and König [14] for the younger age group. Further, it will be possible to highlight differences between younger (mostly working) and older participants who experienced the pandemic from another point of view.

### 2.5. Statistical Analysis

Metric data were not distributed normally and are presented as medians and interquartile ranges. Significant differences between groups were calculated using the Mann–Whitney U test or Kruskal–Wallis test with post hoc Bonferroni correction. Correlations were calculated using Spearman’s correlation coefficient. Nominal and ordinal data are presented as absolute numbers and percentages. Chi-squared tests were used to test for differences and correlations.

Sum scores were calculated for altruism, well-being, resilience, conspiracy theories, trust in media, and trust in institutions as described [21]. For all questions on Likert scales, where a further item, ‘I don’t know’ or ‘I don’t know the decision’, was added, this item was substituted with the mean value of the scale in order to be able to use the whole data set without biasing the results. Further details are provided in [19].

Linear multiple regression was used to explain altruism based on the predictor variables for both age groups. A minimum sample size of 251 for a linear regression model with 13 independent predictors, a type one error of 5%, a power of 95%, and a squared R of 0.1, which was significantly different from 0 (corresponding to a small effect size of 0.11), was calculated using G*Power version 3.1. Regression was controlled for linear relationships between predictors and independent variables, and regression diagnostics were conducted to check for normality and mean 0 of residuals, homoscedasticity, multicollinearity, and autocorrelation of error terms using the Durbin–Watson test, and outliers using DF-Beta statistics, Cook distance, and leverage diagnostics.

*p*-values < 0.001 are indicated with ***, <0.01 with **, <0.05, *, and *p*-values ≥ 0.05 are regarded as not significant (n.s.). All statistical analyses were performed using the SPSS version 27.

## 3. Results

### 3.1. Population Characteristics

Data from 1426 individuals were collected. The group of participants aged 18–69 years included 1169 persons, and the group of participants aged 70 years included 257 individuals. The demographic characteristics of the data set were representative of age, sex, and municipality. The demographic variables “living together with children from 0 to 17 years” and “working in the health sector” could not be analyzed in detail in the group of individuals aged 70 years or more due to the low number of cases in this group.

Overall, the median altruism sum score was 24 (interquartile range [Q1, Q3] = 20, 26) (Appendix A).

#### 3.1.1. Differences between Age Groups

In the age group of 70 years or more, this median score was significantly higher than in the group aged 19–69 years (24 [20;27] vs. 23 [20;26], *p* < 0.05. The two age groups differed in terms of several demographic characteristics (Appendix A). In the group aged 18–69 years old, significantly fewer female participants were found than in the group aged ≥ 70 years (50.3% vs. 57.2%; *p* < 0.05). Educational status was significantly higher in the younger group (*p* < 0.001; for percentages, see Appendix A), as well as having another citizenship (9.4% vs. 3.1%; *p* < 0.001), living in a household only with adult persons who were not COVID-19 patients at risk (40.1% vs. 20.7%; *p* < 0.001), and belonging to the group of persons with COVID-19 infection (18.7% vs. 12.9%; *p* = 0.028). In the group of participants aged 70 years or older, the percentage of those living in a single household was significantly higher (33.1% vs. 12.9%; *p* < 0.001), as was living with a COVID-19 patient at risk (44.9% vs. 15.9%; *p* < 0.001) and suffering from a chronic disease (41.2% vs. 12.1%; *p* < 0.001). The characteristics of the mother tongue and economic situation in the last three months also differed significantly between the younger and older age groups. Participants in the younger age group were less likely to be Italian (25.2% vs. 35.2%; *p* < 0.001) and more likely to speak another or more than one language (8.9% vs. 0.4%; *p* < 0.001). The economic situation of older persons was not significantly worse than that of the younger age group (86.4% vs. 64.2%; *p* < 0.001).

Figure 1 presents the percentage of statements in the five altruism items for the two age groups. The statement ‘I enjoy doing things for others’ was significantly (*p* < 0.01) more agreed to in the age group of participants aged 70 or older, as well as the question ‘Seeing others prosper makes me happy’ (*p* < 0.001). The question ‘I come first and should not have to care so much for others’ found significantly less agreement in the group of persons aged 70 or older (*p* < 0.01). The questions ‘I try to help others, even if they do not help me’ and ‘I really care about the needs of other people’ did not significantly differ between the two age groups and found generally less agreement than the other three questions.

#### 3.1.2. Baseline Characteristics per Age Group

The five questions were summed to obtain a sum score representing overall altruism. Appendix A presents the median and first and third quartiles of altruism for the demographic characteristics of the sample for all participants as well as for the two age groups. Females were significantly more altruistic than males (*p* < 0.001) in both age groups from 18 to 69 years (*p* < 0.001) and in the group of persons aged ≥ 70 years (*p* < 0.05). People with Italian citizenship were significantly less altruistic than those with other citizenship (*p* < 0.01); this difference was significant in the group of participants aged 18–69 (*p* < 0.01), but not in the group of older participants.

In subgroup comparisons, differences between native languages were found for individuals aged 18–69 years (*p* < 0.05), with significantly lower altruism in German-speaking participants than in Italians (*p* < 0.05) or persons speaking another or more languages (*p* < 0.05) without Bonferroni correction. Applying the Bonferroni correction, no significant difference between languages in the younger age group could be detected. No significant differences were detected in the older age groups.

Participants living in a single household were significantly less altruistic than other participants (*p* < 0.05), while participants living in a household with COVID-19 patients at risk were significantly more altruistic (*p* < 0.05). People living in a household with only other full-aged persons who were not at risk were found to be less altruistic than the other participants (*p* < 0.05); this difference remained significant in the subgroup of individuals aged ≥ 70 years (*p* < 0.05). Persons working in the health sector were significantly more altruistic than others (*p* < 0.001), as were those suffering from chronic diseases (*p* < 0.001). Individuals with a chronic disease were significantly more altruistic both in the age group of 18–69 years (*p* < 0.05) and in the age group of ≥70 years (*p* < 0.05). Urban residents were significantly (*p* < 0.05) more altruistic than rural residents, a difference attributable to the level of altruism in the age group 18–69 years (*p* < 0.05), but not in the age group ≥ 70 years.

No significant differences regarding altruism could be found for the variables educational status, economic situation within the last three months, former COVID-19 infection, living in a household with children aged 0 to 17 years, and vaccine hesitancy, neither in the whole sample nor in the two age subgroups.

### 3.2. Associations of Altruism with Predictors of Vaccine Hesitancy

The correlations between altruism and metric predictors of vaccine hesitancy are presented in Table 1. Altruism was significantly positively correlated with age (r = 0.081, *p* < 0.01), trust in institutions (r = 0.209, *p* < 0.001), and agreement with the general national vaccination plans (r = 0.127, *p* < 0.001). Altruism was negatively correlated with well-being (r = −0.090, *p* < 0.01). Significant correlations in the entire study population were also observed between the two age subgroups. Conspiracy thinking, resilience, and trust in the media were not correlated with altruism, either in general or age groups. The correlations between significant predictors are shown in the table. The highest correlation among the independent predictors of vaccine hesitancy was between agreement with the national vaccination plan and trust in institutions in both age groups.

### 3.3. COVID-19 and COVID-19 Vaccination Correlates

Our interest in the effects between altruism and COVID-19 depending correlates was measured using questions on agreement with decision making, general restrictions, adoption of preventive measures, and trust in the COVID-19 vaccination. Spearman’s rank correlation coefficients between altruism and questions on COVID-19 and COVID-19 vaccination measured on a 6-point Likert scale are presented in Appendix A.

#### 3.3.1. Altruism and Agreement with Decision Making

The attributes of the sample regarding SARS-CoV-2 and vaccination, compared with altruism, are given in Appendix A. Altruism was significantly positively correlated with the agreement with decisions regarding COVID-19 (r = 0.148, *p* < 0.001), COVID-19 vaccination (r = 0.160, *p* < 0.001), and compulsory vaccination in general (r = 0.161, *p* < 0.001). The correlation between altruism and the three questions shown in Figure 2 was much stronger in the group of participants aged ≥ 70 years than in the younger age group (r = 0.258, *p* < 0.001; r = 0.269, *p* < 0.001; and r = 0.261, *p* < 0.001, respectively).

#### 3.3.2. Altruism and Support of Actual Restrictions

For the whole group, the following questions about the support of actual restrictions were positively correlated with altruism: mandatory face masks in closed public rooms (r = 0.188, *p* < 0.001), mandatory face masks in the open air (r = 0.134, *p* < 0.001), restricted opening hours for bars and restaurants (r = 0.112, *p* < 0.001), distance rules in closed public rooms (r = 0.183, *p* < 0.001) and in the open air (r = 0.149, *p* < 0.001), restrictions on not individual physical and sports activities (r = 0.057, *p* < 0.05), closure of national (r = 0.071, *p* < 0.05) and regional (r = 0.119, *p* < 0.001) borders, compulsory self-isolation after contact with infected persons (r = 0.218, *p* < 0.001), as much smart working as possible (r = 0.090, *p* < 0.05), lockdown (r = 0.069, *p* < 0.05), prohibition of visiting friends and parents not living in the same household (r = 0.089, *p* < 0.05) and closure of hotels (r = 0.091, *p* < 0.05). Only the limitations of individual physical and sports activities (r = −0.092, *p* < 0.05) were negatively correlated with altruism. No significant altruism effect was found for any form of distance learning or for the closure of communal borders.

In the younger age group, restrictions on non-individual physical and sports activities, closure of national borders, and lockdowns were not significantly correlated with altruism. In the older age group, only one view item was significantly correlated with altruism: restricted opening hours of bars and restaurants (r = 0.125, *p* < 0.05), closure of national borders (r = 0.124, *p* < 0.05), compulsory self-isolation after contact with infected persons (r = 0.126, *p* < 0.05), and lockdowns (r = 0.127, *p* < 0.05).

#### 3.3.3. Altruism and Preventive Measures within the Last Seven Days

All measures adopted within the last seven days to prevent the spread of the virus (despite the use of antibiotics to prevent or treat the virus) were significantly positively correlated with altruism. In the age group of individuals aged ≥ 70 years, the highest correlations with altruism were achieved for preventive measures regarding the use of disinfectants (r = 0.316, *p* < 0.001) and social distancing in public (r = 0.319, *p* < 0.005). In the younger age group, the highest correlation was found for the items ‘handwashing with water and soap for 20 s’ (r = 0.259, *p* < 0.005) and ‘wearing a face mask in the public’ (r = 0.229, *p* < 0.001).

#### 3.3.4. Altruism and COVID-19 Vaccination

Of the statements regarding the COVID-19 vaccine, three were significantly positively correlated with altruism: ‘I believe the vaccine can help to contain the spread of the virus’ (r = 0.063, *p* < 0.05), ‘If the vaccine was recommended for me, I would do it’ (r = 0.105, *p* < 0.001), and ‘If your doctor recommended a COVID-19 vaccination, how likely would you be to get vaccinated?’ (r = 0.090, *p* < 0.01). We found a stronger correlation in the age group of persons aged 70 years or older for all three statements than in the younger age group; the first was not significant in the younger age group. The statements ‘If I knew I was already infected with COVID-19, I would not get the vaccine and ‘If everyone else is vaccinated against COVID-19, then I should not get vaccinated’ were negatively correlated with altruism (r = 0.088, *p* < 0.05 and r = 0.146, *p* < 0.001, respectively), the first one not being significantly correlated with altruism in the older age group.

The necessity and harmfulness of the COVID-19 vaccine were investigated using eight statements. All four statements regarding the necessity of COVID-19 vaccination were negatively correlated with altruism: ‘The vaccine is not effective’ (r = −0.072, *p* < 0.01), ‘Herd immunity is going to be achieved with the spread of the virus’ (r = −0.100, *p* < 0.001), ‘It is just a normal flu/does not exist’ (r = −0.108, *p* < 0.001), and ‘It is just a profit for the pharmaceutical industry’ (r = −0.108, *p* < 0.001). No significant correlations were detected in the younger age group for the statement ‘The vaccine is not effective’ and in the older age group for the statement ‘It is just a normal flu/does not exist’. Three of the four statements regarding the harmfulness of the COVID-19 vaccination are negatively correlated with altruism: ‘Long term risks are not known’ (r = −0.062, *p* < 0.05), ‘New vaccines carry additional risks in the RNA’ (r = −0.111, *p* < 0.01), ‘There are doctors who advise against it’ (r = −0.067, *p* < 0.01). The statement ‘An obligation to vaccinate certain groups with priority will lead to great socio-political discussions’ is not significantly correlated with altruism. In the older age group, the statement on mistrust of long-term risks was not significantly correlated with altruism, whereas in the younger age group, advice from medical doctors was not significantly correlated.

### 3.4. Regression Analysis

A linear regression model to explain altruism using sociodemographic, personal, and pandemic-related factors was calculated for each age group. The results of the multiple linear regressions are shown in Table 2 (unstandardized beta coefficients with 95% confidence intervals and *p*-values).

Dichotomous demographic variables that were significantly correlated with altruism (Appendix A) were included in the model for each age group as well as significantly correlated sum scores and age (Table 2). Appendix A lists the correlations between the different items regarding agreement with restrictions and vaccination. Items with the highest correlations were included in the model for each question group per age group.

In the question group concerning support of actual restrictions, ‘compulsory self-isolation after contact with infected persons’ was included in both models, and ‘lockdown’ only in the model of the older age group. Furthermore, the term ‘limitation of physical or sportive activities’ was used in the model of younger participants, according to Hajek and König [14].

In the group of items regarding actual actions within the last seven days, for the younger age group, we included the item ‘hand washing with water and soap for at least 20 s, ‘and in the older age group, the items ‘disinfect hands, when hand washing is not possible. ‘Wearing a face mask in the public’ was included in both models. In the older age group, the term ‘social distancing in the public’ was highly correlated with ‘wearing a mask in the public’ (r = 0.730, *p* < 0.001); it was not included in the regression model.

Finally, the belief that natural immunity is achieved with the spread of the virus was included in the older age group model.

For younger participants, the model was calculated with and without dummy variables for the mother tongue. No significant effects were found. The results of the model that did not include the mother tongue are presented in detail in the table.

The results of the two linear regression models are presented in Table 2. All independent terms were checked for linearity with altruism. No patterns are observed in the scatter plots.

In the younger age group, regression analysis showed that higher levels of altruism were significantly associated with the female gender (beta = 1.611 [1.100; 2.122]; *p* < 0.001), as well as lower well-being (−0.108 [−0.186; −0.030]; *p* < 0.01), higher trust in institutions (0.058 [0.030; 0.086]; *p* < 0.001), higher support of self-isolation after contact with an infected person (0.263 [0.070; 0.457]; *p* < 0.01), lower support of ‘limitation of individual physical or sportive activities’ (−0.417 [−0.581; −0.253]; *p* < 0.001), and higher agreement with taking the ‘Washing my hands often with water and soap for 20 seconds’ (0.497 [0.321; 0.674]; *p* < 0.001) measure within the last 7 days. Age, citizenship, suffering from a chronic disease, working in the health sector, urban/rural residency, and wearing a face mask in public were not identified as significant predictors for the younger age group.

In the older age group, regression analysis showed that higher levels of altruism were significantly associated with decreasing age (−0.214 [−0.310; −0.118]; *p* < 0.01) and female sex (1.225 [0.094; 2.357]; *p* < 0.05) as well as living not in a household with adults who were not at risk (−1.478 [−2.850; −0.105]; *p* < 0.05), higher trust in institutions (0.180 [0.110; 0.250]; *p* < 0.001), and higher adoption of hand disinfection (0.560 [0.192; 0.928]; *p* < 0.01). The variables suffering from a chronic disease, well-being, ‘Support compulsory self-isolation after contact with a positive person’, ‘Lockdown’, wearing a face mask in public, and the belief that natural herd immunity is reached with the spread of the virus were not significant predictors in the model for older participants.

Regression models were calculated using forward and stepwise selection; the results were the same.

For both age groups, the residuals were normally distributed with a mean of 0, the homoscedasticity assumption was fulfilled, and the maximal Cook’s distance of residuals was 0.015 for the younger age group and 0.0016 for the older age group. The maximum value of leverage points was <0.02 in both groups, and even analysis of the DF-Beta statistics did not reveal any outliers.

The variance inflation factor (VIF) was <1.25 for all predictors in both groups; thus, multicollinearity was not found. Durbin–Watson statistics in SPSS were not calculated for weighted data. Recalculation of the models using the unweighted data did not show any differences in results to the models using weighted data and returned a Durbin–Watson Statistic value of 1.966 for the younger age group and 1.891 for the older age group, which indicates no autocorrelation problems of the residuals.

## 4. Discussion

This study aimed to investigate altruism in the context of the COVID-19 pandemic among the adult population of South Tyrol, Italy. The results confirmed that altruism was higher in the older age groups (≥70 years). Especially the items ‘I enjoy doing things for others’ and ‘seeing others prosper makes me happy’ found higher agreement in the older population, while the item ‘I come first and should not have to do so much for others’ was agreed significantly more often in the younger population. Generally, in both age groups, a positive attitude towards COVID-19 vaccination was significantly associated with higher altruism. Greater agreement with decisions taken by authorities is associated with higher altruism in both age groups. Agreement with actual restrictions is positively associated with altruism, with only one exception in the younger age group: Agreement with restrictions on individual sports is negatively associated with altruism in the younger age group.

Comparing our results in the age group of 18–69 years with those of Hajek and König [14], who reported, for the respective age group, positive associations of altruism with female gender, younger age, chronic diseases, COVID-19 vaccination support, and support of ‘no restriction on individual sports activity’, most correlations are confirmed by the current study. ‘Having children’ in our results was not associated with altruism.

Regression analysis identified the female gender and support for sports activities as significant positive predictors of altruism. While the female gender is widely known for its association with altruism, support for individual sports activity is an interesting novel effect that is worth investigating in more detail, since sports activity can have both prosocial and antisocial aspects [34,35].

Further predictors identified in the younger age group were lower well-being, which was only slightly (Kandall’s Tau-b = 0.060, *p* < 0.05) associated with having a chronic disease, trust in institutions, compulsory self-isolation, and handwashing. Thus, the results confirm that general positive support for pandemic measures and trust in authorities are positively associated with altruism. In the younger age group, age could not be identified as a significant predictor in the regression model, while in the older age group, lower age predicted altruism as well as the female gender. However, living in a household with adults not at risk of COVID-19 was a negative predictor. This corresponds to the findings of Jones et al. [16], in which altruism was more associated with close relatives than with other vulnerable people.

Mask-wearing, social distancing, and handwashing are positively associated with altruism. For both age groups, hand washing and disinfection were identified as positive predictors of altruism. Compulsory self-isolation was a significant predictor in the younger age groups. The motivation for wearing a mask changes with age, with older people wearing it for themselves, whereas younger people wear the mask even for altruistic reasons [36]. The present study confirms that personal protective behavior is related to altruism in an age-specific manner, in that altruism is a weaker predictor of wearing a face mask in the context of COVID-19. However, with increasing age, the motivation to wear a face mask may change from altruism to egoism. This is suggested by the correlation coefficient of altruism with the question of whether wearing a face mask in public is supported. While we found a significant positive correlation in the younger age group, this correlation was not confirmed in the older age group.

Finally, in the regression model for both age groups, vaccination terms were not found to be predictors. Generally, the correlation coefficients between altruism and vaccine items in the questionnaire were smaller than those between altruism and other pandemic restrictions or measures.

Daily prosocial behavior is an important factor in altruism [37]. In our survey, the older age group consisted of persons who had already retired and met different needs during the COVID-19 pandemic compared to the younger age group. A questionnaire regarding altruism in the elderly was developed, especially for subjects aged ≥ 70 years [27]. This questionnaire is used to quantify altruism. Thus, a valid and reliable measure of altruism is used to explain the interactions between age and gender. Older persons living alone, together with children, or with COVID patients at risk have been identified as more altruistic than those living with adults who are not at risk. Furthermore, younger age and female gender were significant positive predictors. The only significant prosocial factor in the regression model was “disinfection of hands.”

For the younger age group, we were able to compare the results to a similar study investigating the adult age group up to 69 years using a different altruism questionnaire [14]. Since most results are concordant, we additionally confirm that persons continuing outdoor activities and sports during pandemic lockdowns may not be antisocial, but are still considered altruistic.

A video intervention with prosocial and altruistic content was able to significantly improve COVID-19 vaccination behavior more in young than in older study participants [38]. This suggests that targeted interventions may increase altruism in an age-dependent manner and give differential importance to altruism to improve pandemic preparedness and vaccination behavior. Given the confirmed predictive value of altruism for positive pandemic-like behavior in younger adults and the theoretical possibility of raising altruism through behavioral intervention in this age group, our findings indirectly support further interventional studies aimed at increasing altruism.

Sociodemographic and individual vaccine hesitancy attitudes predicted vaccine uptake during the COVID-19 pandemic; however, the high correlations between altruism and metric attitudes of vaccine hesitancy reported here and in the literature [39] indicate that addressing individual vaccine hesitancy beliefs may not lead to behavioral change, as other hesitancy beliefs may continue to impede vaccine uptake. Altruism and the social responsibility of protecting others have often been invoked in the medical consultations of people with vaccine-hesitant behavior [40]. Altruism has been used repeatedly in health communication strategies, but it is not clear whether this has actually increased vaccination willingness [41]. The success of this communication strategy with a greater willingness to vaccinate has been described in the context of the closest social environment for one’s own family [18,42,43]. Given the differential responses of older and younger age groups to the associations between altruism and pandemic-related attitudes, the use of altruism in health communication strategies might also be more age sensitive.

Limitations: First, the overall age was not a significant predictor. Since the altruism questionnaire used was developed for older persons, we suggest that it is worth repeating the investigation using a possibly more adequate questionnaire for the younger age group. Second, representative information on vaccination intention is important for an agile regional healthcare response to vaccination [44]. In Italy, SARS-CoV-2 and mandatory non-coronavirus vaccination rates are among the lowest in South Tyrol, the northernmost province of the country. This is the first time altruism has been characterized in the general population of South Tyrol. A direct comparison with different Italian regions or other countries is not possible, because there are no established reference values for altruism. Third, the limitations of this study include unmeasured variables of potential importance for vaccination behavior, such as political and religious orientation or the use of alternative and complementary medicine, which are known predictors [45]. Finally, adolescents who could play a significant role in pandemic preparedness were excluded from the survey [46].

## 5. Conclusions

The results confirm that general positive support for pandemic measures and trust in authorities’ information and decisions are positively associated with altruism. Although altruism has also been targeted in pandemic management communications and to improve the uptake of vaccination offers, data on the effectiveness of this strategy during the recent pandemic are lacking. The different and differentiated correlations between the behaviors of the population and the degree of altruistic behavior depending on the age of the target groups, as described here, have generally not been considered. The initial results provide a principal opportunity to address the different effects of altruism on pandemic attitudes in individual age groups to a greater extent in the future.

The association between altruism and pandemic behavior and traits is significant because it suggests that individuals who are more altruistic are more likely to engage in behaviors that protect themselves and others from the spread of infectious diseases. The identified positive and negative correlations between altruism and pandemic attitudes also suggest that age plays a modifying role in the effects of altruism on pandemic preparedness. This means that the relationship between altruism and pandemic behaviors may be stronger or weaker depending on a person’s age. This information can be used to develop targeted interventions to enhance altruism and improve pandemic preparedness in specific age groups.

Overall, the significance of these associations is that they provide insights into how to promote behaviors that can prevent the spread of infectious diseases during pandemics. By understanding the factors that influence pandemic behavior, public health officials and policymakers can develop more effective strategies to protect public health and mitigate the impact of future pandemics. Whether possible interventions to increase altruistic behavior are also influenced in their effectiveness by age remains to be seen.

## Figures and Tables

**Figure 1 behavsci-13-00188-f001:**
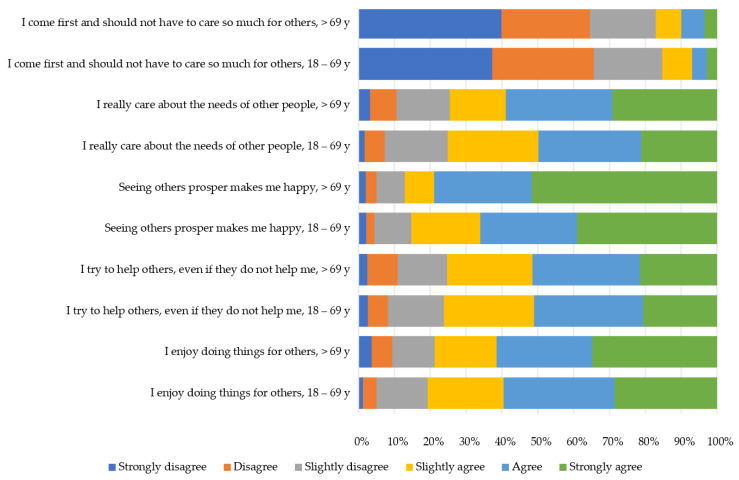
Differences in response patterns to single items of the altruism scoring system by survey participant age groups 18–69 years (*n* = 1169) and 70 years and above (*n* = 257), respectively. Abbreviation: y, years.

**Figure 2 behavsci-13-00188-f002:**
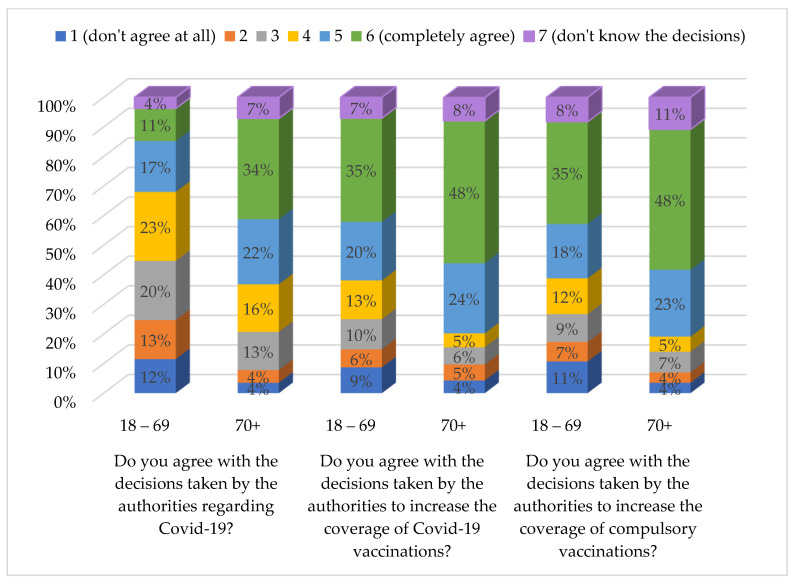
Responses to survey items regarding agreement with the decisions taken by the authorities concerning COVID-19 and vaccinations by age groups 18 to 69 years (18–69) and 70 years and above (70+).

**Table 1 behavsci-13-00188-t001:** Spearman‘s correlation coefficients between altruism, age, and well-being, and between altruism and conspiracy thinking, resilience, or trust in media.

	Altruism	Age	Well-Being	Trust in Institutions	Agree with the National Vaccination Plan
Total					
Altruism	1				
Age	0.081 **	1			
Well-being	−0.090 **	−0.104 ***	1		
Trust in institutions	0.209 ***	0.149 ***	−0.107 ***	1	
Agree with the national vaccination plan	0.127 ***	0.169 ***	n.s.	0.410 ***	1
Age 18–69 years					
Altruism	1				
Age	0.069 *	1			
Well-being	−0.071 *	−0.152	1		
Trust in institutions	0.173 ***	n.s.	n.s.	1	
Agree with the national vaccination plan	0.095 **	0.110 ***	n.s.	0.423 ***	1
Age 70 years and above					
Altruism	1				
Age	−0.238***	1			
Well-being	−0.136 ***	0.148 *	1		
Trust in institutions	n.s.	n.s.	−0.208 **	1	
Agree with the national vaccination plan	0.233 ***	n.s.	n.s.	−0.234 ***	1

*p*-values < 0.001 are indicated with ***, <0.01 with **, <0.05 with *, and *p*-values ≥ 0.05 are regarded as not significant (n.s.).

**Table 2 behavsci-13-00188-t002:** Predictors of altruism in South Tyrol, Italy, in March 2021 in multivariate regression analyses for age groups 18–69 years and 70 years or older, respectively.

Variable	Age 18–69 Years *n* = 1169 R^2^ = 0.126			Age 70 Years or Older *n* = 257 R^2^ = 0.244		
	Beta Coefficient	[95% CI]	*p*-Value	Beta Coefficient	[95% CI]	*p*-Value
Constant term	17.876	[16.038; 19.714]	<0.001	30.752	[22.588; 38.915]	<0.001
Gender	1.611	[1.100; 2.122]	<0.001	1.225	[0.094; 2.357]	<0.05
Age			n.s.	−0.214	[−0.310; −0.118]	<0.01
Citizenship			n.s.	---	---	---
Suffering from a chronic disease			n.s.			n.s.
Working in the health sector			n.s.	---	---	---
Urban/rural			n.s.	---	---	---
Living in a household with adult persons not at COVID-19 risk	---	---	---	−1.478	[−2.850; −0.105]	<0.05
Well-being	−0.108	[−0.186; −0.030]	<0.01			n.s.
Trust in institutions	0.058	[0.030; 0.086]	<0.001	0.180	[0.110; 0.250]	<0.001
Support compulsory self-isolation after contact with a positive person	0.263	[0.070; 0.457]	<0.01			n.s.
Restricted sports activities	−0.417	[−0.581; −0.253]	<0.001	---	---	---
Lockdown	---	---	---			n.s.
Hand washing	0.497	[0.321; 0.674]	<0.001	---	---	---
Disinfection of hands	---	---	---	0.560	[0.192; 0.928]	<0.01
Wearing a face mask in the public			n.s.			n.s.
Natural herd immunity is achieved with the spread of the virus	---	---	---			n.s.

CI, confidence interval; n.s., not significant.

## Data Availability

The data presented in this study are available upon request from the corresponding author. The data were not publicly available because the survey was conducted by the statistical office of the regional administrative authority and had politically sensitive content.

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
