# Peer review of "Age-Related Associations of Altruism with Attitudes towards COVID-19 and Vaccination: A Representative Survey in the North of Italy"

_behavsci, 2023, doi:10.3390/bs13020188_

Round 1

Reviewer 1 Report

Main Notes:

1. 2. The title requires simplification - the second phrase seems redundant.

2. The abstract is too extensive - it contains a lot of result content that is relevant to the conclusions. I propose to write them in a more general way - without presenting numbers.

3. It is worth considering changing the justification for choosing the region for research - South Tyrol. Geographical factors do not matter much here, while covid morbidity rates.

4. The methodological section lacks the aim of the research, the research problem and the (main) and/or detailed and specific indication of the methods. The authors only indicate the research technique and tool, avoiding including them in quantitative research (in the survey method using the survey technique) - these issues should be specified.

5. I propose to refer more broadly to classical theories on altruism. It is worth reading the texts: Batson, C. D., & Ahmad N. Y. (2009), Altruism and Prosocial Behavior in Groups; Batson, C.D., & Powell, A.A. (2003). Altruism and pro-social behavior. In T. Million & M. J. Lerner & I. B. Weiner (Ed.), Handbook of Psychology; R. Kałużny (2021) Pro-Social and Altruistic Behaviors of Military Students in Random Events and other publications on altruistic behavior.

Reviewer 2 Report

From a research point of view, the article is interesting. The authors correctly described the research methods. Population characteristics have been correctly described. The same applies to items such as: Differences Between Age Groups etc. The research results were presented in an interesting way. I also have no objections to Discussion and Conclusion. I believe the article can be published.

Author Response

We thank the reviewer to the response. No changes were requested.

Reviewer 3 Report

The work presented here is well organized however it still needs some minor modifications 

the authors did not mention about their hypothesis clearly and no clear statement for the research question 

the authors need to add more interpretations to address the significance of their results 
